# Protocol for a randomised controlled trial to evaluate the effectiveness of the 'Care for Stroke' intervention in India: a smartphone-enabled, carer-supported, educational intervention for management of disabilities following stroke

K Sureshkumar,[1,2] G V S Murthy,[1,2] Hannah Kuper[2]

[1]Public Health Foundation of India, Indian institute of Public Health - Hyderabad, South Asia Centre for Disability, Inclusive Development and Research (SACDIR), Hyderabad, Telangana, India
[2]Department of Clinical Research, International Centre for Evidence in Disability, London School of Hygiene and Tropical Medicine, London, UK

**Correspondence to**
K Sureshkumar;
suresh.kumar@iiphh.org

## ABSTRACT

**Introduction** The rising prevalence of stroke and stroke-related disability witnessed globally over the past decades may cause an overwhelming demand for rehabilitation services. This situation is of concern for low-income and middle-income countries like India where the resources for rehabilitation are often limited. Recently, a smartphone-enabled carer-supported educational intervention for management of physical disabilities following stroke was developed in India. It was found to be feasible and acceptable, but evidence of effectiveness is lacking. Hence, as a step forward, this study intends to evaluate clinical effectiveness of the intervention through a randomised controlled trial.

**Methods** The objective of the study is to evaluate whether the 'Care for Stroke' intervention is clinically and cost-effective for the reduction of dependency in activities of daily living among stroke survivors in an Indian setting. This study is designed as a randomised controlled trial comparing people who received the intervention to those receiving standard care. The trial will be pragmatic and outcome assessor-blinded. The primary outcome for the study is dependency in daily living measured by the Modified Rankin Scale (MRS). A total of 266 adult stroke survivors who fulfil the eligibility criteria will be randomised to receive either 'Care for Stroke' intervention or standard treatment and will be followed up for 6 weeks. The main analyses will compare participants allocated to the 'Care for Stroke' intervention versus those allocated to the standard treatment group on an 'intention-to-treat' basis, irrespective of whether the participants received the treatment allocated or not. The dichotomised MRS scores (0–3 and 4–6) in both the groups will be used to calculate the effect estimates with a measure of precision (95% CI) and presented in the results of the trial.

**Ethics and dissemination** The Indian Institute of Public Health-Hyderabad/Public Health Foundation of India – Independent Institutional Ethics Committee and the Ethics Committee of the London School of Hygiene and Tropical Medicine. Dissemination will be through peer-reviewed publications.

## Strengths and limitations of the study

► Effectiveness of the intervention will be established through a randomised controlled trial.
► The trial protocol was pilot tested and was found feasible.
► This is the first ever stroke trial in India evaluating a mHealth rehabilitation intervention.
► Stringent inclusion criteria for participant recruitment.
► The duration of follow-up in the trial is not long.

**Trial registration number** Clinical Trial Registry of India CTRI/2017/07/009014.

## INTRODUCTION

Globally around 15 million people suffer from stroke each year and a quarter of them experience permanent disability.[1] Much of this burden is borne by low-income and middle-income countries (LMICs).[2] The increase in prevalence of stroke[3] and consequently of stroke-related disability may cause an overwhelming demand for rehabilitation services worldwide.[3] This situation is especially of concern for LMICs like India where the resources for rehabilitation are often limited.[3]

Stroke is one of the leading causes of death and disability in India. Given the paucity of data on stroke in India, a systematic review of population-based studies on stroke in India was conducted. Studies included in this review showed that the crude stroke prevalence during the past two decades in India ranged from 44/100 000 persons to 559/100 000 persons,[4] and the cumulative incidence of stroke in India ranged from

105–152/100 000 person per year.[4] These estimates on stroke incidence and prevalence are found to be higher than those reported from high-income countries.[5] The growing burden of stroke-related disability and the unmet need for rehabilitation following stroke in India poses a major public health challenge.

There is a paucity of global evidence on the effectiveness of therapy-based stroke rehabilitation, especially in long-term care.[6 7] Available evidence shows that there is no single physical rehabilitation approach that is more effective than combinations of care.[8] Provision of information to stroke survivors and caregivers has been shown to improve functional outcomes.[9] However, the best way to do this is still unclear. Recently, mHealth options are rising substantially and mobile technology has been substantially used to communicate for health-related reasons. Though mHealth strategies have developed various solutions to meet the needs of stroke survivors, the best way to use this approach in stroke rehabilitation is also still unclear.[10] There is insufficient evidence for tele-rehabilitation services.[11] This context provides a strong grounding for the development of cost-effective multidimensional stroke rehabilitation interventions to meet the demands of the stroke survivors. In the absence of organised stroke care services and with the limited resources available for rehabilitation, a comprehensive approach to address the growing burden of stroke-related disability in India becomes pertinent.[12] This approach could be pivotal in integrating various strategies for rehabilitation[3] (educational, community-based rehabilitation, digital technology, self/supported management and so on). It could also be useful for targeting the full range of impacts of stroke, including on impairments, activity limitations and participation restriction, as outlined in the 'biopsychosocial conceptualisation of disability framework' for the intervention, as proposed by the International Classification of Functioning Disability and Health (ICF).[13]

A smartphone-enabled carer-supported educational intervention was developed by our group for the management of physical disabilities following stroke in India.[14] This intervention was named as 'Care for Stroke'. It was developed using the systematic approach to development and evaluation of complex interventions, as recommended by the Medical Research Council (MRC) in the UK.[15 16] We intended to bridge the gaps in access to stroke services through this innovative intervention which optimises relevant public health practice with the support from mobile devices such as smartphones, personal digital assistants and other wireless devices.[17] To the best of our knowledge, there is no other stroke rehabilitation intervention enabled through mHealth platforms that are available and relevant to India.

The intervention was evaluated for its feasibility and acceptability in an Indian context.[18] The intervention includes information about stroke and the ways to manage physical disability following stroke. It contains a practical demonstration of functional poststroke exercises to acquire the functional abilities necessary to perform everyday tasks, adaptive techniques to perform one's own daily activities independently and a specific section on assistive devices that could enable participation of the stroke survivors in their daily tasks.[14] Findings from the pilot-testing showed that the 'Care for Stroke' intervention was feasible and acceptable in the Indian context.[18] About 95% of the stroke survivors and all the caregivers (100%) rated the intervention as 'excellent', based on its (a) overall credibility, (b) feasibility and (c) user-friendliness.[18]

However, feasibility and acceptability alone will not be sufficient to inform implementation and scalability.[16] Nor will it be enough in order to advocate for change in policy towards implementation of an intervention.[19] Investigating the intervention clinical and cost-effectiveness will provide insights for planning, implementation and the potential scalability of the intervention, especially in countries with limited resources. Given the methodological quality of the available evidence,[9–11] there is a pressing need to conduct a rigorous (randomised, controlled, sufficiently powered) clinical trial to demonstrate the effectiveness of the 'Care for Stroke' intervention.

## OBJECTIVE

The objective of the randomised controlled trial is to evaluate whether the 'Care for Stroke' intervention is effective for the reduction of dependency in activities of daily living among stroke survivors compared with people receiving standard treatment in an India setting. The primary outcome for the study is disability measured by the modified Rankin Scale (MRS).

## METHODS
### Overview

This trial will be a pragmatic, randomised, outcome assessor-blinded trial to quantify the effectiveness of the Care for Stroke intervention on reducing dependency in activities of daily living following stroke. A total of 266 adult stroke survivors who fulfil the eligibility criteria will be randomised to receive either 'Care for Stroke' intervention or standard treatment and will be followed for 6 weeks. The flow chart of the entire trial process is provided in figure 1.

### Pragmatic design and the uncertainty principle

The effectiveness of the intervention in routine practice can be assessed using the pragmatic trial design. Until now, there is no evidence for effectiveness of stroke rehabilitation interventions that is unidisciplinary, led by a physician, neurologist or a physiotherapist alone.[12] However, a physiotherapist or physician-driven unidisciplinary rehabilitation is what is commonly practised in the context of stroke rehabilitation in India.[12] Given the lack of evidence, there is a natural uncertainty among the health professionals involved in provision of stroke care

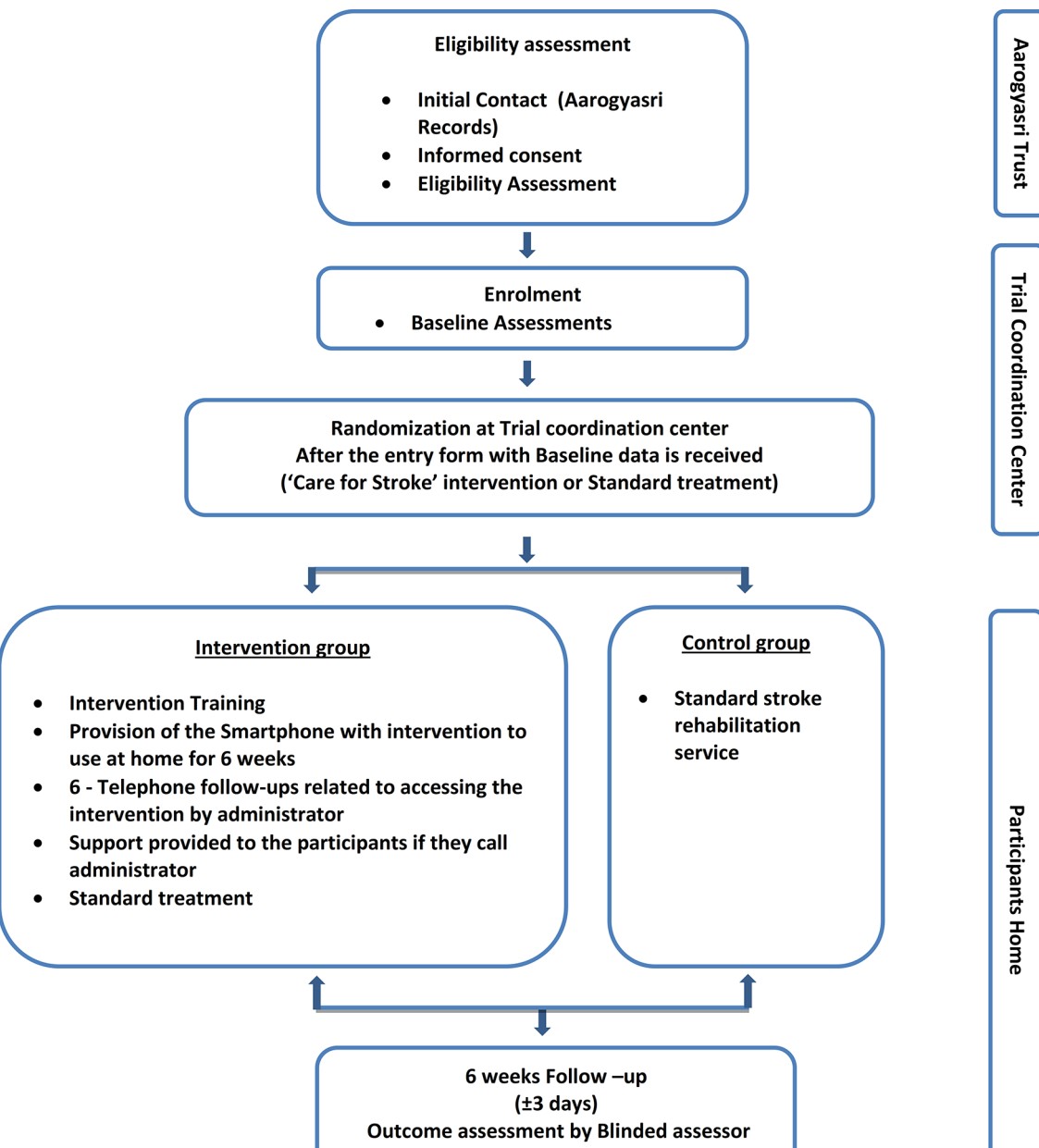

**Figure 1** Flow chart of the Care for Stroke trial.

about what intervention could work best for the stroke survivors in an Indian context. The eligibility for participant recruitment in the 'Care for Stroke Intervention' trial will be based on this uncertainty principle. This approach to assess participant eligibility is well established.[20]

## SETTING
### Participant recruitment
Participants will be identified using their contact details from treatment records for their first ever stroke. These details for stroke survivors in India exist in two places. Participant diagnosis and details can be collected from the hospital records from which an individual received treatment for his/her stroke. It is also available at the government health insurance department where the cost of the

treatment for stroke is covered by this insurance department. Hence, participants will be identified through both these options. The identified participants will be contacted at their home for consent and recruitment. The intervention will be provided to the participants at home and they will be asked to use the intervention in their home.

## ELIGIBLE PARTICIPANTS
### Inclusion criteria
► Adults (aged ≥18 years).
► Recent diagnosis of first ever stroke as defined by the WHO.[21]
► Any level of stroke severity (score 1–42, according to National Institute of Health (NIH) stroke scale.[22 23])

► Stroke survivor medically stable (reaching a point in medical treatment where life-threatening problems following stroke have been brought under control).

► Poststroke functional status of the stroke survivor: requiring assistance of at least one person to perform daily activities such as transfers, self-care and mobility (ie, scoring less than the maximum score obtainable in one or more components of the Barthel Index[24]).

► Stroke survivor residing with a primary caregiver (family member) at home.

## EXCLUSION CRITERIA

► Severe cognitive difficulties (scoring >1 in orientation, executive function, inattention and language components of the NIH Stroke Scale for cognition[25]).

► Severe communication problem (scoring >1 in dysarthria and best language component of the NIH Stroke Scale[22 23]).

► Stroke survivor functionally dependent because of other pre-existing conditions (eg, amputation, fracture and dementia).

► Stroke survivor without a primary caregiver.

► Stroke survivor unwilling/unable to adhere to the study protocol.

► Stroke survivors who did not meet the training requirements regarding operation of a smartphone. This criterion was deliberately placed just to make sure that there is no dropout after the recruitment. It was based on the observations from previous piloting.

## RANDOMISATION

Stroke survivors will receive all-usual treatment for stroke. Participants eligible for inclusion will be identified by a trial investigator. The eligible participants will be initially contacted by telephone and they will be visited in person at their home by the investigator to share the details about the study to the participant and the identified caregiver. A participant information sheet outlining the study objectives, risks and benefits along with brief information sheet about stroke will be provided to the participant. Written informed consent for participation in the intervention will be sought from all participants or from the next of kin if the participant is unable to consent.

An entry form will be used to collect baseline information including the contact details of the participant and the identified caregiver. This information will be forwarded to the independent randomisation centre and the participants eligible for inclusion will be randomised to the intervention or control arm in a 1:1 ratio using a secure, central, password-protected, web-based system. The intervention will be started within 24 hours of randomisation.

## SAMPLE SIZE ESTIMATION

The two main factors that determine the number of participants needed in this trial are the estimated event rate and the size of the treatment effect. The primary outcome for

the 'Care for Stroke' trial is dependency in activities of daily living measured at 6 weeks postrecruitment.

Estimated event rate: in a meta-analysis of early supported discharge trial among participants with stroke, 50% of the stroke survivors were either dead or dependent at the end of follow-up and the beneficial effect of the intervention in the treatment group was an odds reduction of 21% of death and dependency.[26]

As a non-inferiority one-sided trial, to evaluate the effectiveness of the smartphone-enabled educational intervention on dependency, there will be a requirement of 266 participants (133 in each group) to detect a 20% difference in dependency among the participants between the treatment groups with 90% power at the 5% level of statistical significance and with 20% loss to follow-up.

A non-inferiority trial could exclude the possibility of a small degree of inferiority of a new intervention relative to an active control given the sample size. The results of the trial provided by the CI will allow concrete evaluation of the precision actually achieved, superseding any power calculation carried out before the starting the trial.

## INTERVENTION

The 'Care for Stroke' intervention will be delivered through a smartphone and it will include information about stroke and the ways to manage poststroke disabilities. The intervention includes 2–3 min of 60 videos in vernacular language organised in five sections. The sections are: (1) information about stroke, (2) home-based exercises, (3) functional skills training, (4) activities of daily living, and (5) assistive devices. The intervention will be self-directed, with participants seeking information in the different categories as they require. The intervention will also have an option for the stroke survivor or the identified caregiver to contact the intervention provider for any technical support in accessing the intervention through smartphone.

## INTERVENTION ARM

The stroke survivor and their caregiver will receive 45–60 min of training on accessing and use of the intervention (watching videos) via the smartphone. Participants will then be provided with a smartphone preloaded with the 'Care for Stroke' intervention and asked to try it out on their own. Three or more errorless attempts to retrieve any required part of the intervention from the smartphone will be considered successful training. After successful training, participants will be provided with a smartphone loaded with the intervention and will be asked to use this intervention at their discretion at home over a 6-week period.

The identified caregivers of stroke survivors will be asked to support the stroke survivors as and when necessary to access the intervention from the smartphone. The participants will be telephonically supported at least once in a week during the intervention period. The telephonic

support is essentially to remind and obtain updates from the participants or identified caregivers on utilisation of the intervention. A summary of this conversation will be documented and the notes will be kept privately in a locked cupboard. The participants in the intervention arm will not be restricted from receiving standard treatment for their stroke.

## CONTROL ARM

Participants in the control arm will receive standard post-stroke rehabilitation services. In general, the standard treatment may include provision of physiotherapy (45 to 60 min) at home or in a clinic facility for the stroke survivors based on goals set by the specific therapist or a rehabilitation team.

## OUTCOME MEASURES
### Primary outcome

The primary outcome measure is dependency in activities of daily living and will be measured by the MRS[27] at baseline and at 6 weeks after randomisation. The MRS scale measures the degree of disability or dependence in the activities of daily living of people who have suffered a stroke in six categories. The scores range from 0 (no symptoms) to a maximum of 6 (dead). A dichotomous approach to outcome analysis will be used. Participants' scores will be categorised into MRS scores of 0–3 and 4–6.

## SECONDARY OUTCOME

Secondary outcome measures will be:
▶ Modified Barthel Index.[24]
▶ Modified Caregiver Strain Index.[28]
▶ Quality of Life measured by World Health Organization - Quality of Life Brief (WHOQOL–BREF).[29]
▶ Use of healthcare and rehabilitation services (therapy, hospitalisation and medication, AYUSH, traditional practices and so on).
  This information will be collected through questionnaire at baseline and after 6 weeks. The smartphone application has an inbuilt monitoring mechanism where the usage of the intervention by the participants will be tracked.

Costs for rehabilitative care will be collected from participants both in the treatment groups to see whether the Care for Stroke intervention delivered through a smartphone reduces the overall costs of care (cost-effectiveness).
▶ Direct costs of healthcare and rehabilitation since the time of stroke.
▶ Indirect costs (a family member giving up paid employment and taking the role of a caregiver, travel costs and so on).

## FOLLOW-UP

An outcome form will be completed at 6 weeks after randomisation or at death, if either happens sooner. A blinded outcome assessor will evaluate all the outcomes (primary and secondary) at baseline and at 6 weeks. A relatively short follow-up period has been selected as The Stroke Therapy Academic Industry Round Table strongly recommends a shorter follow-up period to reduce variation in clinical outcome that could occur due to subsequent stroke events that are unrelated to the trial.[24] This will also allow accurate assessment of the outcome.[30]

## ADVERSE EVENTS

Adverse events are very common among acute stroke survivors. Some of the expected adverse events during the trial are as follow:
1. Death due to any vascular causes (eg, myocardial infarction, recurrent stroke).
2. Hospitalisation due to poststroke complications such as infections, brain oedema, seizures, deep vein thrombosis, urinary tract infections, pressure sores and shoulder subluxation, dislocation and fracture.
3. Occurrence of secondary stroke.

These events will be documented during follow-up telephone calls and it will be presented to an independent data safety and monitoring committee for unblinded review.

## DATA COLLECTION AND MANAGEMENT

This trial will be centrally coordinated from the Trial Coordination Center (TCC) at the Indian Institute of Public Health (IIPH) Hyderabad. Baseline data will be collected by the investigator and follow-up data will be collected with appropriate translation by an independent blinded outcome assessor on paper forms. These data will be securely scanned and sent to the TCC for entry into the password-protected, secured electronic database. An independent data safety and monitoring committee (DSMC) will be set up to monitor data collection and management. A trial steering committee will also be set up to oversee the conduct of the trial.

## ANALYSIS

The main analyses will compare all those allocated to the 'Care for Stroke' intervention with those allocated to the standard treatment group on an 'intention-to-treat' basis, irrespective of whether the participants received the treatment allocated or not. The imbalance in recruiting equal number of participants if any will be addressed during the analysis phase using appropriate statistical techniques. The dichotomised MRS scores (0–3 and 4–6) in both the groups will be used to calculate the effect estimates with a measure of precision (95% CI) and presented in the results of the trial. Subgroup analysis for the primary outcome will be based on stroke severity, location of the

participant (urban/rural), gender and age at stroke. Interaction tests will also be used to test whether the effect of treatment (if any) differs across these subgroups.

## Recruitment of participants

The trial will identify and recruit participants from the hospital records as well as stroke insurance records available at the Aarogyasri trust until the sample size is achieved. Currently, the average stroke insurance claim rate through this trust is 10–12 stroke survivors per month. Hence, it would take approximately 32–36 months for recruiting the proposed number of participants in this trial.

## PATIENT AND PUBLIC INVOLVEMENT

Patients and public were not involved for the purpose of protocol development.

**Acknowledgements** The authors thank The Wellcome-trust-DBT India Alliance for funding the research study. They thank the independent institutional research ethics committee of the PHFI - Indian Institute of Public Health Hyderabad for granting scientific and ethics approval to conduct this research study. They also thank the consultants from Suchir softech and Selva photography for developing the software application and digitisation of the content of the intervention.

**Contributors** KS conceived, designed and drafted the manuscript. GVSM and HK played a crucial role in conception of the research study and provided substantial guidance in designing and conducting evaluation.

**Funding** This work was supported by the Wellcome Trust/DBT India Alliance Fellowship [grant number **IA/CPHE/16/1/502650** ] awarded to Dr Sureshkumar K .

**Competing interests** None declared.

**Patient consent** Not required.

**Ethics approval** Ethical approval for this trial has been obtained from the independent institutional research ethics committee at the public health foundation of India (IIPH) Hyderabad. Results of this trial will be published in relevant, peer-reviewed, indexed, international journal.

**Provenance and peer review** Not commissioned; externally peer reviewed.

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
