## [Reviewer comments · BMJ Open]

ARTICLE DETAILS

TITLE (PROVISIONAL)	Protocol for a randomised controlled trial to evaluate the effectiveness of the 'care for stroke' intervention in India; a smartphone-enabled, carer-supported, educational intervention for management of disabilities following stroke
AUTHORS	Sureshkumar, K; Murthy, GVS; Kuper, Hannah

VERSION 1 – REVIEW

REVIEWER	Anne Forster University of Leeds, UK
REVIEW RETURNED	23-Oct-2017

GENERAL COMMENTS	Thank you for a well written protocol. I appreciate that the study might already have started but I have one particular concern and one query. Concern: It is stated that: participants' are excluded if they "Did not meet the training requirements regarding operation of a smartphone". Unless I have misunderstood this particularly criteria is not applied until after randomisation when: " Three or more errorless attempts to retrieve any required part of the intervention from the smartphone will be considered successful training." If this is the procedure than it would seem there is a potential for differential dropouts between the two groups? Perhaps the paper requires greater clarification or the procedures need amending. Query: on the Trials Registration website it states that outcomes will be assessed at 3 and 6 months rather than the 6 weeks reported in this protocol. 6 weeks does seem a short follow-up time. In places the author speaks in the first person "I believe that non-inferiority trials.... " Page 13. I am not sure whether BMJ Open has a policy on this?
--

REVIEWER	Susan Mahon AUT Univeristy, New Zealand
REVIEW RETURNED	10-Nov-2017

GENERAL COMMENTS	 1. The abstract is not clear enough and the aims are not well defined. There is also a number of grammatical errors. Study outcomes should be defined. 2. The first sentence in the introduction cites a 2004 reference. There are a number of recent Global Burden of Disease studies which would be more appropriate to use. 3. Reference 4 does not seem a valid reference, there are a couple of more recent studies in terms of epidemiology in India (Kamalakaman, 2016; Bhata, 2014). 4. The focus should be on primary and secondary prevention? 5. How many people with stroke in India have access to rehabilitation, and what specific rehabilitation is offered? what is the average length of stay, what is offered in the community? This needs to be addressed to support why you need the intervention.... 6. Biopsychosocial conceptualization of disability framework' for the intervention, as proposed by the ICF is mentioned yet there is no reference (page 8) 7. Page 8 mentions Mhealth yet there is no definition of that this means 8. Page 9 discusses findings from the pilot study yet there is no mention of sample size in either group. 9. The objective should start include how you are going to reduce dependency ...measured by what? Needs another sentence 10. Pragmatic design and the uncertainty principle....it is hard to know what this means? This is a clinical trial and should follow consort guidelines ? There is no consort flowchart 11. The type of stroke is not mentioned in the eligibility criteria...are all stroke types included? 12. Surely having access to a smart phone must be part of the eligibility criteria? 13. Of concern is how the research team have access to a patients contact information ? This is not described. 14. Te sample size estimate is not well described , and no peer reviewed evidence to support the calculations..... First tense is used in both paragraphs which should be removed. 15. Primary outcome poorly described and not sufficient. Is it a shift in mRS which is being measured and what is it? If this is your primary measure it needs to be alot clearer 17. There is no discussion on secondary measures in the methods and how they will be administered...by telephone or face to face 18. the method for determining costs for rehabilitation is insufficient 19. A conclusion is not needed for a study protocol
--

REVIEWER	Dr. Fred Stephen Sarfo Kwame Nkrumah University of Science & Technology Ghana
REVIEW RETURNED	19-Nov-2017

GENERAL COMMENTS	 1. The time for recruitment of 32-36 months is quite long and authors should consider including other study sites to abbreviate the trial recruitment duration. 2. Description of the intervention, such as duration of session of physical exercises, are not clearly defined. Subjects assigned to the intervention are to use it at their own discretion should be reviewed. How many sessions per week should they have? Such information would be helpful to the authors when analyzing outcomes. 3. Author should kindly review the manuscript for syntax and grammar errors
--

VERSION 1 – AUTHOR RESPONSE

Responses for editor and reviewers comments

Editors Comments	Responses	Changes in the Manuscript
Please revise the 'Strengths and limitations' section of your manuscript.	Strengths and Limitations in the manuscript are now revised.	Page 5 - highlighted in yellow
please include a copy of the SPIRIT checklist	TIDieR Check list was previously included but we have added SPIRIT checklist now	Supplementary file – SPIRIT Checklist
Reviewer -1 Comments	Responses	Changes in the Manuscript
There is a potential for differential dropouts between the two groups?	The authors thank the reviewer for the concern. The reviewer was correct. The particular criterion (errorless attempts) was not applied until after randomisation. This criterion was deliberately placed just to make sure that there is no dropout after the recruitment. It was based on the observations from previous piloting. We also intend to employ minimisation techniques and also address the imbalance if any during the analysis phase using appropriate statistical techniques.	None.
6 weeks does seem a short follow-up time.	Care for Stroke is an educational intervention and hence in a RCT of this nature, it is important to have a short follow-up time to identify the exact component that influenced the treatment effect rather than having a long follow-up time which might contaminate the intervention and produce biased results. There are several studies which mention this phenomenon. It was also based on our observations from pilot. Hence we took this decision. We intend to document and transparently report the difference between the trial protocol and the actual conduct will be in the future.	None.
Reviewer - 2 Comments	Responses	Changes in the Manuscript
The abstract is not clear enough and the aims are not well defined. There are also a number of grammatical errors. Study outcomes should be defined.	The abstract section is now revised as advised.	Page 3 - highlighted in yellow

The first sentence in the introduction cites a 2004 reference. There are a number of recent Global Burden of Disease studies which would be more appropriate to use.	Recent Article – 2017 is cited now.	Page 20 - highlighted in yellow
Reference 4 does not seem a valid reference, there are a couple of more recent studies in terms of epidemiology in India (Kamalakaman, 2016; Bhata, 2014).	Recent Article – 2017 is cited now.	Page 20 - highlighted in yellow
The focus should be on primary and secondary prevention?	There has been several global research studies and evidence on primary and secondary prevention of stroke and very little has been researched on tertiary prevention. There is paucity of evidence related to this level of prevention for stroke especially in low and middle income countries and hence, we intend to focus on tertiary stroke prevention and care.	None
How many people with stroke in India have access to rehabilitation, and what specific rehabilitation is offered? What is the average length of stay, what is offered in the community? This needs to be addressed to support why you need the intervention....	The authors thank the reviewer for comment. However, there have been no studies and publications relevant to the details the reviewer has mentioned. We have mentioned our work related to it and also cited some of our publications related to this in our manuscript.	Page 6 and 7 - highlighted in yellow
Biopsychosocial conceptualization of disability framework' for the intervention, as proposed by the ICF is mentioned yet there is no reference (page 8)	We have cited the reference for ICF	Page 7 - highlighted in yellow
Page 8 mentions Mhealth yet there is no definition of that this means	mHealth is a component of eHealth. To date, no standardized definition of mHealth has been established. For the purposes of the trial we have used the definition from, the Global Observatory for eHealth (GOe).	Page 7 and 22 - highlighted in yellow
Page 9 discusses findings	The pilot study was not a trial and we had	None

from the pilot study yet there is no mention of sample size in either group	only one group purposively sampled for it. It focussed on the feasibility and acceptability of the intervention. There was a publication related to this and the reference was cited this in the manuscript.	
The objective should start include how you are going to reduce dependency ...measured by what? Needs another sentence	As advised, we have specified the primary outcome and the tool that will be used for its evaluation.	Page 9 - highlighted in yellow
Pragmatic design and the uncertainty principle...it is hard to know what this means? This is a clinical trial and should follow consort guidelines? There is no consort flowchart	We have very clearly mentioned what uncertainty principle means to this trial in the manuscript. We also had a TIDieR checklist for our initial submission. As advised by the editor, We will follow the guidelines of SPIRIT and provide the SPIRIT checklist.	Supplementary file
The type of stroke is not mentioned in the eligibility criteria...are all stroke types included?	Yes all the Stroke types are included. We have mentioned it in the eligibility criteria. WHO definition includes both Ischaemic and Haemorrhagic stroke.	Page 11 - highlighted in yellow
Surely having access to a smart phone must be part of the eligibility criteria?	The answer is No. Because the participants will be provided with a smartphone in the trial.	Page 15 - highlighted in yellow
Of concern is how the research team have access to patients contact information? This is not described	We have mentioned this clearly in Setting section. We will obtain the details from the hospital records and the records of government health insurance programme for stroke.	Page 11 – highlighted in yellow
The sample size estimate is not well described, and no peer reviewed evidence to support the calculations..... First tense is used in both paragraphs which should be removed.	The authors feel that they have explained about the sample size estimation in detail. There is a peer reviewed evidence (systematic review) to support the calculation. First tense statements have been removed and rephrased as advised.	Page 14 highlighted in yellow
Primary outcome poorly described and not sufficient. Is it a shift in MRS which is being measured and what is it? If this is your primary measure it needs to be a lot clearer	The authors thank the reviewer for this comment. The focus is not on the shift in MRS. It is on dichotomised MRS. These changes are made in the manuscript. Especially the sample size.	Page 16 highlighted in yellow.
There is no discussion on secondary measures in the	It will be face to face assessment conducted by a blinded outcome assessor.	Page 17 – 18 highlighted in

methods and how they will be administered...by telephone or face to face	This is mentioned in the manuscript.	yellow
A conclusion is not needed for a study protocol	We agree with the reviewer.	Conclusion removed
Reviewer - 3 Comments	Responses	Changes in the Manuscript
The time for recruitment of 32-36 months is quite long and authors should consider including other study sites to abbreviate the trial recruitment duration.	We thank the reviewer for the comment. The time for recruitment was planned based on the pilot and it also helps to ensure sufficient time is taken to conduct the trial. However, the investigators have already approached several recruitment sites and we are open to the inclusion of additional sites.	None
Description of the intervention, such as duration of session of physical exercises, is not clearly defined. Subjects assigned to the intervention are to use it at their own discretion should be reviewed. How many sessions per week should they have? Such information would be helpful to the authors when analysing outcomes.	Care for stroke is an educational intervention and not a therapy-based intervention hence information about exercises and other aspects are included as videos for sensitizing the stroke survivors and enhancing their knowledge and hence the purpose of the intervention is only patient education and its impact on dependency thus the participants can use it at their discretion. Telephonic follow-up further reiterate the information in the intervention.	None
Author should kindly review the manuscript for syntax and grammar errors	The revised manuscript has been reviewed and proof read by a native English speaker.	Complete Manuscript

VERSION 2 – REVIEW

REVIEWER	Susan Mahon AUT University New Zealand
REVIEW RETURNED	13-Feb-2018

GENERAL COMMENTS	The authors have taken into account the recommendations from the previous review.
---

REVIEWER	Anne Forster University of Leeds, United Kingdom
REVIEW RETURNED	01-Mar-2018

GENERAL COMMENTS	I think the authors should address the issue that they acknowledge, of potential for differential dropouts, as an inclusion/exclusion criteria (below) is applied after randomisation.  Stroke survivors who did not meet the training requirements regarding operation of a smartphone I am not sure that these participants should be excluded. But the authors need to be clearer about the approach they are taking (as in their response to comments). Please compare the analysis section in the abstract and that in the main section of the paper to ensure they are saying the same thing.
---

VERSION 2 – AUTHOR RESPONSE

Responses for editor and reviewers comments on Revision 1

Editors Comments	Responses	Changes in the Manuscript
Please revise the strengths and Limitations section of your Manuscript	The Strengths and Limitations section of the manuscript is now revised.	Page 5 - highlighted in yellow
Please ensure Protocol reports all aspects of information as included in the registry – Especially Sample size in the methods section	The methods section – especially the calculation of sample size for the trial is now revised. We will also revise the protocol submitted to the trial registry as advised.	Page 3 and 14 - highlighted in yellow
Reviewer - 2 Comments	Responses	Changes in the Manuscript
None	None	None
Reviewer - 1 Comments	Responses	Changes in the Manuscript
Participant exclusion – Author need to be clearer about the approach they are taking (as in their response to comments)	The Responses related to this comment from the reviewer is now included in the manuscript at Exclusion criteria section and Analysis section	Page 12 and 19 - highlighted in yellow
Ensure Analysis Section in the abstract and in the main section say the same thing	The authors have made sure that the analysis section in the abstract and the main section say the same.	Page 3 and 19- highlighted in yellow